# Effectiveness of 38% Silver Diamine Fluoride Application along with Atraumatic Restorative Treatment for Arresting Caries in Permanent Teeth When Compared to Atraumatic Restorative Treatment in Adults—Study Protocol for a Randomized Controlled Trial

**DOI:** 10.3390/mps5060087

**Published:** 2022-10-26

**Authors:** Anju Varughese, Chandrashekar Janakiram, Balagopal Varma, Anil Mathew, Shankar Rengasamy Venugopalan, Venkitachalam Ramanarayanan, Prabath Singh

**Affiliations:** 1Department of Conservative Dentistry and Endodontics, Amrita School of Dentistry, Amrita Vishwa Vidyapeetham, Cochin 682041, India; 2Department of Public Health Dentistry, Amrita School of Dentistry, Amrita Vishwa Vidyapeetham, Cochin 682041, India; 3Department of Pedodontics and Preventive Dentistry, Amrita School of Dentistry, Amrita Vishwa Vidyapeetham, Cochin 682041, India; 4Department of Prosthodontics, Amrita School of Dentistry, Amrita Vishwa Vidyapeetham, Cochin 682041, India; 5College of Dentistry and Dental Clinics, Department of Orthodontics, The University of Iowa, 801 Newton Road, Iowa City, IA 52242, USA

**Keywords:** atraumatic restorative treatment, caries arrest, dental caries, silver diamine fluoride, survival of the restoration

## Abstract

Introduction: Dental caries in the adult population that require preventive and therapeutic treatment are generally neglected in rural communities. The determination of the effectiveness of the application of 38% silver diamine fluoride (SDF) in arresting caries lesions when combined with atraumatic restorative treatment (ART) is very important, as it serves as a preventive and restorative procedure to regain the function of the permanent dentition. The assessment of optimal SDF application with ART, in comparison with ART alone, in managing cavitated carious lesions in a pragmatic setting, is the need of the hour to recommend optimal dental care, especially in rural settings which have minimal access to comprehensive dental care. Methods and Analysis: The clinical trial will enrol 220 adults (18–65 years) with cavitated carious lesions attending the Amrita School of Dentistry in the Ernakulam district, India. This study is a randomized, controlled trial with a 1:1 allocation ratio in two parallel groups. Study arm 1 will receive 38% SDF application and ART, and study arm 2 will receive ART only. A digital radiograph will be taken immediately after restoration (baseline) as well as at the end of the 6th month for evaluation of caries arrest. The assessment of the survival of the restoration will be done on the 7th day, 30th day, and at the end of the 6th month. The final analysis would include both the tooth and person levels. Ethics and Dissemination: This trial adheres to the principles of the Declaration of Helsinki and the guidelines of the Indian Council of Medical Research (ICMR). This study protocol has been approved by the Institutional Review Board. This trial has been registered prospectively with the Clinical Trial Registry of India (Registration No: CTRI/2021/12/038816).

## 1. Introduction

Oral health is essential to overall health and well-being. The global prevalence of oral diseases is estimated to be almost 3.5 million people worldwide. The lack of availability, accessibility, and affordability of basic oral health care is a significant problem in low- and middle-income countries, particularly amongst rural populations [1]. The sequelae of typical untreated oral diseases are commonly excruciating. They can constitute unrelieved discomfort, debilitated quality of life, and shortened work efficacy. Oral diseases impose a high financial crisis on families and the medical system due to treatment costs incurred [2]. Since the early appearance of dental caries is mostly asymptomatic, most of the population refrains from seeking oral care, which can lead to painful events in the future.

According to the World Health Organization, the population of people above the age of 65 years is increasing at a rate of 2.5% annually. This shift in the age distribution of the population will be markedly experienced in developed, as well as lesser-developed countries, by 2050. Poor oral health is deleterious to the holistic state of well-being [3]. Dental caries is the least treated disease among older adults. The prevalence of dental caries is found to be 50–84% among the adult population of India [4,5]. Early treatment of dental caries with the advent of new techniques and novel materials enables minimally invasive and less expensive treatment today. Preventive interventions are the most effective treatment against dental caries. The most frequently used preventive interventions are the application of pit and fissure sealants and fluoride varnish [6]. Currently, the state of the art of caries treatment is represented by resin-based composite restorations either in anterior or posterior teeth [7,8,9].

Atraumatic restorative treatment (ART) is a highly successful community-based approach for the treatment or secondary level of prevention of dental caries. It is an alternative approach for managing dental caries which involves the removal of infected tooth tissue using conventional hand instruments alone, without the use of anesthesia and rotary equipment. One of the indications for the pertinent use of ART concerned the elderly, particularly those who are impaired. The first such study on atraumatic restorative treatment was carried out amongst elderly subjects who were housebound due to psychiatric, physical, or psychological problems. After 1 year of placement, 79% of the ART restorations were considered successful [10]. In a two-year randomized controlled trial, ART presented the probability of survival rates of 84% compared to conventional techniques, consisting of the preparation of (the decayed) tooth using a rotary instrument (for removal of infected tissues) under local anesthesia, followed by resin-modified glass ionomer cement restoration [11]. When the effect of ART on adult patients’ comfort was assessed, it was noted that ART caused less anxiety than the traditional restorative methods using rotary instruments as the vibration and sound of the rotary instruments were absent [12].

ART is an important component of the Basic Package of Oral Care (BPOC) which also includes Affordable Fluoride Toothpaste (AFT), and Oral Urgent Treatment (OUT). This has been recommended by the WHO, and has been a successful community-based approach to reducing the burden of dental caries [13]_._ Considering the global rise of the elderly population with natural dentition in the coming decennium, a study covering the effectiveness of the ART approach, as part of a package of oral care for use among the adult population, should receive serious attention. However, the ART approach is not successful in some cases, especially in cavities that are less accessible through conventional hand instruments. Improper removal of caries lesions may lead to bacterial invasion, secondary caries, and failure of restoration [14].

Advanced dental caries preventive technology comprises treatment with various remineralizing agents. Silver diamine fluoride (SDF), introduced in 1969, has been used for the management of dentin hypersensitivity and to arrest dental caries. The Central Pharmaceutical Council of Japan’s Ministry of Health and Welfare accepts a 38% SDF solution as a dental treatment for caries [15]. SDF (38% *w*/*v* Ag (NH_3_)2F, 30% *w*/*w*) is a colourless topical agent. It is composed of 24.4–28.8% (*w*/*v*) silver, 5.0-5.9% fluoride, and ammonia at pH 10 [16]. A layer of the silver-protein conjugate is formed when silver diamine fluoride is topically applied to a decayed tooth. This enhances resistance to acid effect and digestion by enzymes. Silver diamine fluoride outperforms other anticaries medicaments as the silver and fluoride ions permeate 25 microns into the enamel and about 50–200 microns into dentinal tubules. The fluoride ions advance remineralization, and silver ions have effective antimicrobial action. The application of SDF is simple, painless, and non-invasive [17].

Remarkable results were noted in elderly patients when SDF was applied annually for the prevention and arrest of occlusal surface caries as well as root caries [18]. Rosenblatt et al. recommend SDF application as a more effective caries preventive intervention than fluoride varnish and observe it to be an effective, efficient, safe, and affordable caries-preventive agent that meets the WHO Millennium Goals criteria for 21st-century medical care [19]. Application of 38% SDF solution is recommended over the use of 5% sodium fluoride varnish to better arrest cavitated carious lesions on the coronal tooth surface of permanent dentition [20]. The current clinical trial reports on SDF application support its effectiveness in preventing and cessation of radicular caries, deep coronal lesions remineralization, along with hypersensitive dentin treatment [21,22].

However, one of the noted drawbacks of SDF application is its potential to cause discoloration of the teeth and oral tissues [17]. Black staining is an often-reported side effect of SDF application and influences the acceptability of this therapy. The use of potassium iodide (KI) directly after SDF application results in masking the staining of the affected dentine [23]. Applying KI over SDF creates an insoluble layer of yellow precipitate of silver iodide, thus obscuring the black staining on the tooth [24]. The effectiveness of SDF in preventing caries progression was not affected by the application of KI [25]. The application of saturated potassium iodide solution to reduce discoloration was not found to be effective in altering or preventing caries in a study among community-dwelling senior citizens [26].

Due to their safety, efficiency, feasibility, and effectiveness in dental caries management, ART and SDF are employed in routine treatments for all age groups. Yet if the initial prevention strategies fail, there is a need for effective management to arrest the disease, restore the cavitated tooth, and reduce the negative impact of caries lesions. The silver modified atraumatic restorative technique, known as the SMART technique, is a modification of ART, as stated in a case report, wherein the application of SDF is followed by conditioning with polyacrylic acid and restoration of the cavitated tooth with GIC [27]. By using SMART with chemically sealed restoration, the nutrient source for any persistent bacteria is cut off, thus killing the microbes, arresting and remineralizing the caries dentin, preserving the remaining tooth structure, and enhancing the pulp vitality [27]. The application of an antimicrobial agent like SDF before the placement of a restoration proved to be more effective, slowing the progression of the carious lesion [28]. There are no reports which suggest hampering of shear bond strength of glass ionomer cement to carious dentine by the application of KI solution after SDF placement [29,30].

According to the socioeconomic census in India in 2011, 68% of the population is still living in rural areas [31] where the possibility of health systems integrating to their fullest extent is still a difficult task. Therefore, easily accessible and less expensive treatments should be promoted for changing the population’s attitude towards dentistry, although India has been advancing in various aspects of oral health [32]. Educating on new concepts that can be easily perceived and applicable through community health workers and dental auxiliaries in society can bring about a drastic change in this prejudice about dental well-being. With the advancement of community-based procedures such as atraumatic restorative treatment and silver diamine fluoride application more attention can be bought to the population towards dental care and its importance.

A combination of SDF application and ART together would be an innovative, minimally invasive method that imparts both bactericidal effects leading to caries arrest and restores the function of the tooth. Thus, we would be able to better address the drawbacks of both these procedures when applied individually. Such a combination of 38% silver diamine fluoride application along with atraumatic restorative treatment on the cavitated carious lesion of permanent teeth in adults has not been reported in the literature. If found successful, this would be an extremely valuable technique that will be of great advantage for oral health in community-based projects.

The objective of the clinical trial is to compare the effect of the application of 38% SDF along with atraumatic restorative treatment, against atraumatic restorative treatment alone, in arresting active occlusal carious lesions on permanent teeth. It also intends to compare the survival of the restoration. The trial also assesses the patient satisfaction and acceptability of the application of the SDF.

## 2. Methods

### 2.1. Study Design

This is a randomized, controlled trial with two parallel comparator arms, having a 1:1 allocation ratio (Figure 1).

### 2.2. Study Settings

The trial will be conducted at Amrita School of Dentistry, Amrita Institute of Medical Sciences, Kochi, India.

### 2.3. Trial Registration

The trial protocol has been registered with the Clinical Trial Registry of India at http://ctri.nic.in accessed on 1 November 2021. (Registration No: CTRI/2021/12/038816 dated 21 December 2021, Registered on 2 November 2021). Recruitment of study participants will begin in August 2022 and the trial is anticipated to conclude by December 2024. This clinical trial protocol has been developed following the SPIRIT 2013 Statement (https://www.spirit-statement.org/ accessed on 10 August 2020).

### 2.4. Study Population

Any individual between 18 to 65 years with one or more cavitated carious lesions in the occlusal and/or proximal surface of the permanent posterior teeth.

### 2.5. Eligibility Criteria

#### 2.5.1. Inclusion Criteria

Male or female individuals, between the age of 18 to 65 years, with either one of the following: Class 1 caries (only occlusal surface involved in posterior teeth) or Class 2 caries (occlusal and proximal surface of posterior teeth), without involving the pulp.Patients understand the significance of SDF and ART for the treatment and prevention of tooth decay.Patients are willing to attend the follow-up visits as required for the trial.Patients will be enrolled irrespective of special needs if they are willing and able to comply with all the trial requirements.Patients with the absence of periodontal disease.

#### 2.5.2. Exclusion Criteria

Patient with stomatitis or ulcerative gingivitis.Developmental abnormalities of enamel and dentin.Serious non-communicable medical conditions include cardiac failure and poor glycaemic control.Cavitated teeth missing greater than a third of the coronal tooth structure or having pulpal involvement. Pulp-exposed tooth, obvious discoloration, premature hypermobility, or presence of a periapical abscess or a sinus will be regarded as a tooth with pulpal involvement.Known silver allergy.During the intervention, if there is inadvertent pulp exposure, the patient will be excluded from the study.

### 2.6. Definition of Study Condition (Dental Caries)

A cavitated dental caries is defined as a carious lesion with loss of tooth surface integrity instead of loss of enamel and exposure of the underlying dentin. If left untreated, the carious lesion is likely to progress.

According to the modified International Caries Detection and Assessment System (ICDAS), the subject teeth surface with stages 5 and 6 is defined as cavitated lesions [33]. A visual examination aided by tactile detection using a community periodontal index (CPI) probe under a dental chair light source would be used to assess the status of dentine caries. The tooth surface will be dried. The caries lesions may be obvious on simple visual clinical examination. Visible caries will be confirmed by probing gently over the entire coronal surface of the tooth. Yellowish-brown rough-walled cavities will be termed active lesions if they are easily penetrated by the CPI probe. The occlusal, buccal, lingual, mesial, and distal will be assessed in each posterior tooth. An intra-oral periapical radiograph (IOPAR) will be done before the start of the study to assess the extent of the caries lesion and the pulpal status of the tooth and to confirm the absence of pulp involvement.

### 2.7. Interventions

After patient selection, the treatment will be given to the patient according to the selection process. The study participants will be allocated to either one of two parallel arms.

Study arm 1: will receive SDF application and ART restoration.

Study arm 2: will receive ART restoration.

All treatment will be done in a dental chair.

At the baseline visit, an assessment of cavitated caries lesions will be done on all the study participants. The ADA Caries Risk Assessment Form (Age > 6) would be used to assess the risk. Depending on the study arm, ART with or without SDF will be applied to selected teeth (Figure 1). The demographic features of participants and their caries status will be recorded.


*Procedure to be followed for study arm 1:*


A preoperative photograph of the selected tooth will be taken. The caries lesion entrance will be widened and with help of a dental hatchet, the thin unsupported frangible enamel will be removed. In the next step, the decayed tissue will be excavated with a spoon excavator, first at the dentin enamel junction and thereafter from cavity walls and the pulpal floor. The cavity will then be rinsed with water and air-dried, followed by isolation with cotton rolls.


*Application of Silver Diamine Fluoride*


Using a disposable micro applicator tip, the SDF (e-SDF, 38% silver diamine fluoride, Kids-e-dental LLP, Globus Medisys, Mumbai, India) will be applied to the walls of the prepared cavity for 10 s and allowed to air dry for one minute. One drop of 38% SDF solution (2.24 F-ion mg/dose) would be required per tooth. A saturated solution of potassium iodide (Lugol’s Iodine solution 5%, Bio balance, Pali Marwar, Rajasthan, India) will be applied with a different microsponge to decrease color change and will be repeated 1-3 times until no further black discoloration is observed [17].


*Placement of ART restoration*


Glass ionomer cement (Glass Ionomer High Strength Posterior restorative, GC Corporation, Tokyo, Japan) will be prepared according to the manufacturer’s instructions. The restorative material will be placed into the prepared cavity using a plastic filling instrument. The press finger technique will be used for approximately one minute and the restoration will be coated with petroleum jelly. A carver would be used to remove the excess material. The height of restoration will be adjusted after the bite is checked using articulation paper. A photograph and a digital radiograph of the restored tooth will be taken after the procedure and at the end of the study.


*Procedure to be followed for study arm 2:*



*Placement of ART restoration*


A preoperative photograph of the selected tooth will be taken. The preparation of the cavity will be done as stated above using hand instruments. Conditioning of the prepared tooth surface will be done with an applicator tip saturated with the polyacrylic acid liquid for 10–15 s. Glass ionomer cement (GIC Fuji IX, G C CORPORATION, 76-1 Hasunuma-Cho, Tokyo 174-8585, Japan) will be prepared according to the manufacturer’s instructions and placed into the cavity using a plastic filling instrument. The press finger technique will be followed, and the surface of the restoration will be coated with petroleum jelly. The bite will be checked using articulation paper and the occlusal adjustments will be done. A photograph and a digital radiograph of the restored tooth will be taken after the procedure and at the end of the study.

### 2.8. Primary Outcomes

#### 2.8.1. Assessment of Survival of the Restoration 

The follow-up will be conducted on the 7th and 30th days, and the end of the 6th month to evaluate the survival of the restoration (Table 1).

Each restoration will be assessed according to codes and criteria used in ART studies as proposed by Jo E Frencken [13]. Cavities treated with SDF and ART will be considered arrested caries lesions when the tooth surfaces with active caries at baseline visit change into surfaces with arrested caries beneath an intact restoration. The restored tooth is asymptomatic and clinically functional.

In the SDF and ART or ART-treated tooth, if caries progress beyond the dentine, it will be considered not arrested. In such conditions, the patients will be advised to follow standard treatment protocols. During the follow-up period, if the restored tooth is extracted due to infection, trauma, or fracture, the concerned tooth will not be included in the study.

#### 2.8.2. Radiographic Evaluation of Caries Arrest of the Lesion

At the end of the 6th month, a digital radiograph evaluation of the surviving restoration will be done for comparison with the baseline radiograph. Radiographic evaluation will be done only at the end of the 6th month as it takes a minimum of 3–4 months for radiographic changes to be evident.

The digital radiographic evaluation of the restored teeth will be done by two independent blinded examiners. The efficacy of the treatment will be assessed by the presence or absence of an increasing radiolucent area below the restoration. Each radiograph will be classified using Ekstrand criteria [34].

### 2.9. Secondary Outcome

A questionnaire will be given to the participants after each visit to assess self-reported satisfaction and acceptability of the treatment. The discomfort of treatment experienced by the patient will be assessed using the facial scale of Wong–Baker [35].

### 2.10. Participation Timeline

Each eligible individual selected for the trial will be participating in the trial for a period of 6 months from their first visit (baseline). The participation timeline: (t_−1_ and t_0_) baseline visit [either first visit (t_−1_) for Enrolment and/or second visit (t_0_) for Allocation], (t_1_) revisit on the 7th day after intervention, (t_2_) revisit on the 30th day, and (t_3_) revisit at the end of 6th month (Table 1).

### 2.11. Sample Size

The sample size was determined by using the G*Power statistical analysis program. With an anticipated effect size of 0.4, the probability of alpha error of 0.05, and 80% power, the sample size is estimated to be 100 per group. Expecting a 10% attrition in the follow-up visit, the final sample size per group is 110. Therefore, the total sample size is 220.

### 2.12. Study Implementation

The adult patients visiting for routine dental care to dental school will be recruited for the study. A clinical examination will be carried out on prospective participants. If the individual satisfies the inclusion criteria, the participant will be informed of the options for caries prevention by ART ± SDF. If they agree, they would be offered the choice of participation in the trial and the consenting process will be followed. The patient screening will continue till the target population is attained (n = 220).

Enrolment of the study population will be done over 24 months and follow-up visits are expected to be completed at the end of the 30th month, which will be followed by analysis and report writing. The total study duration is of three years.

The selected participants for the trial will be randomly allocated to a study arm, as per computer-generated randomization. The randomization code will not be revealed to the principal investigator until the participant has been recruited into the clinical trial after all the baseline assessments are complete. This is to ensure allocation concealment. Opaque-sealed envelopes containing randomization codes, in compliance with the randomization list, will be employed for allocation concealment. The randomization will be conducted by the research assistant. The randomization list will remain with the trial coordinator for the whole duration of the research. Due to the apparent nature of the treatment involved, neither the study participant nor the principal investigator (PI) can be blinded.

The PI will be trained for the study requirements, standardized in the assessment of the coronal carious lesion, caries cessation, 38% SDF application and ART, or ART alone, eliciting the required details from the study subjects in a consistent replicable method, and counseling for compliance. The procedure to be conducted and the data to be collected at each study visit will be inspected in detail. During the training sessions, details about procuring and entering the quality data in the clinical record forms will also be discussed. Calibration of PI (AV) will be done before and during the study by another clinical expert (CJ) and reliability will be expressed using Cohen’s kappa statistic.

Once an individual is enrolled in the trial, an appropriate effort will be taken to follow the study participant for the entire study period of six months. Each patient will have the right to withdraw from the trial at any time. If it is due to an adverse event, review visits will be arranged by the PI until the adverse event has been resolved. There will be no restrictions in case the trial participant requires any dental treatments other than that of the concerned tooth.

### 2.13. Compliance with Trial Treatment

The follow-up visits are important for the successful completion of this trial. Compliance with clinical visits at designated intervals will be ensured by either a telephonic reminder call or a text message to participants. The appointments may be rescheduled two days earlier or later if the participant wishes to do so or inadvertently misses an appointment. Non-compliance will be recorded if the participant fails to report for two rescheduled appointments, and the participant will be withdrawn from the trial. However, if the subject completes two revisits and thereafter becomes non-compliant, the person-time of participation in the clinical trial will be included in the analysis.

### 2.14. Adverse Events

38% SDF solution and glass ionomer cement, employed in regular dental practice for standard dental procedures, are the agents used for this trial. There are minimal potential risks for the study participants, such as the risk for the patients receiving oral care in a routine clinical setting. The most expected adverse effect is an allergic reaction to SDF. All grades of adverse events occurring to the participants during the treatment and follow-up period will be recorded and reported. The principal investigator would obtain information about adverse events with the help of standard questions and an oral examination at every planned visit of the study participant.

The participants may experience an unpleasant metallic taste. Silver diamine fluoride can cause a “transient pigmentation” of the skin, buccal mucosa, or tooth surface. This is not harmful and resolves itself with exfoliation of skin and mucosa within 2 weeks. Universal precautions would be taken to prevent exposures. A cotton roll isolation of the concerned tooth and application of cocoa butter would be carried out to protect surrounding oral tissues from accidental exposure to SDF.

## 3. Close-Out Procedures

The date of the final visit (6 months from the baseline) of the last randomized participant would be the end of the trial date. Regardless of timing and circumstances, the close-out of the trial will proceed in two stages: analysis and documentation of study results, debriefing of participants, and dissemination of study results.

## 4. Statistical Methods

The observed data will be coded, tabulated, and analyzed using Statistical Package for Social Sciences (version 20). We will assess the outcomes at two levels (the Tooth level and the Individual level). The Chi-square test will be executed to assess the bivariate association between the outcome and sociodemographic variables. Kaplan–Meier survival analysis will be conducted on the censored data for the survival of the restoration. The difference between survival curves will be assessed by the Mandel—Haenszel log-rank test. Factors affecting survival rates will be assessed using the Cox-proportionality regression method. Between the two groups, the arrest of active carious lesion and survival of the restoration will be compared on the 7th day, 30thday, and at the end of the 6th month of the study visit. If there is a variation in outcome in patients with multiple restorations, tooth level analysis will be carried out.

Subgroup analysis will be performed at the tooth level (type of cavity; class1 and class 2, maxillary and mandibular teeth, premolar and molar)

We intend to have two sets of analyses.

The intention-to-treat analysis (including all randomized patients irrespective of whether they have or have not received the prescribed treatment).The per-protocol analysis.

## 5. Analysis of Population and Missing Data

The reasons for participant withdrawal would be qualitatively compared for each randomization group and reported. The effect of missing data on the results will be assessed using sensitivity analysis of the augmented data sets. The individual dropouts will be included in the analysis, which would be carried out by multiple imputation methods for missing data.

## 6. Patient and Public Involvement

The dissemination of appropriate dental care concepts through community health workers in an easily understandable way can bring a drastic change in the general perception of oral health. This can be further achieved through the advancement of community-based procedures such as atraumatic restorative treatment, a non-aerosol generating procedure, and the application of silver diamine fluoride. For arresting the progression of caries and restoration of the function of a decayed tooth, a combination of 38% silver diamine fluoride along with atraumatic restorative treatment would be a new minimally invasive method, which can be carried out using simple hand instruments. This combination of ART and SDF may be an effective method to prevent and treat dentinal caries in the adult population, especially among those with disability and poor general health.

The study design (randomized controlled trial) will provide evidence of caries arrest following a novel combination technique of restoration with silver diamine fluoride along with atraumatic restorative treatment on the cavitated tooth. The assessment of optimal SDF application with ART, in comparison with ART alone, in managing cavitated carious lesions in a pragmatic setting, is the need of the hour to recommend optimal dental care, especially in rural settings which have minimal access to comprehensive dental care. This trial is designed to address this knowledge gap. If the trial is successful, the impact would be a change in the standard of care and provide greater access to a non-invasive, simple, and inexpensive strategy to manage cavitated caries lesions, thereby helping to reduce pain and associated dental problems in the adult population.

## 7. Ethics and Dissemination

This trial adheres to the principles of the Declaration of Helsinki and the guidelines of the Indian Council of Medical Research (ICMR). The study protocol has been approved by the Institutional Ethics Committee (IEC) of Amrita Institute of Medical Sciences, AIMS, Kochi, India (IEC-AIMS-2021-DENT-280). Following the completion of the trial, an end-of-trial notification and final report would be submitted to IEC, AIMS.

Informed consent would be obtained from the participants by the principal investigator. A printed information sheet and a copy of the consent in the patient’s native language/English would be provided to the participants with sufficient time for them to discuss the trial and decide on their participation. The participant would be made aware of their responsibilities and their right to withdraw participation from trial at any time without prejudice to future care and without citing any reason whatsoever. Any data that breaches the blind will not be presented until the completion of the trial, in order to maintain the integrity of the study. The informed consent form indicates that individual patients’ data may be used for publication purposes. During and upon completion of the study, only the study investigators will have access to the data. Three years after the completion of the trial, de-identified data will be shared with a data archive source. The relevant study results and actionable points will be shared with concerned health authorities.

## Figures and Tables

**Figure 1 mps-05-00087-f001:**
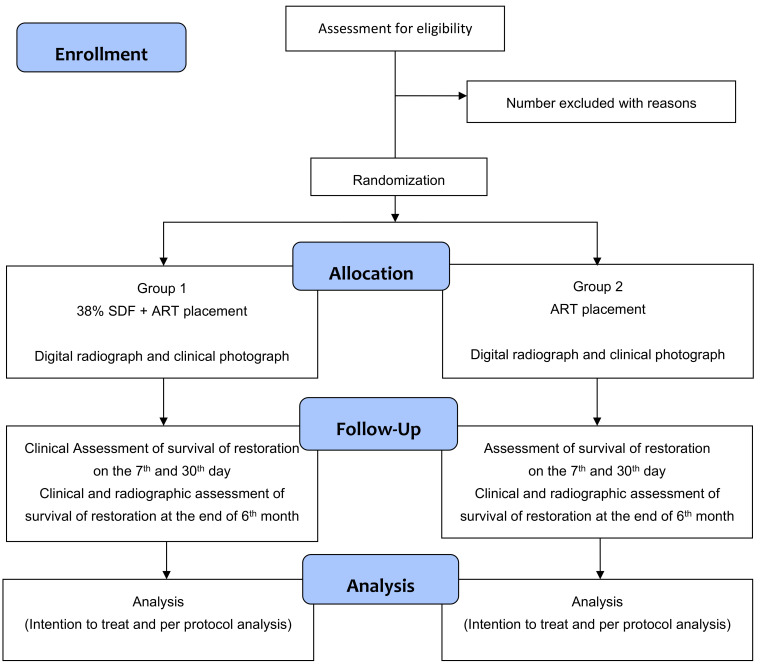
Study flow diagram.

**Table 1 mps-05-00087-t001:** Participant timeline.

	Enrolment	Allocation	Post Allocation	Close Out
Time Point	t-1	t0	t1(7 days)	t2(30 days)	t3(6 months)	tx
*Enrolment*
Eligibility screen	X					
Informed consent	X					
Allocation		X				
Caries excavation		X				
*Interventions*
38% SDF + ART		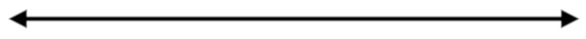	
ART		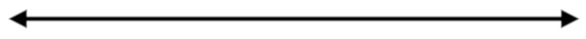	
*Assessments*
Baseline variables cavitated carious lesion	X		X	X	X	
Caries arrest			X	X	X	X
Survival of restoration			X	X	X	X

## Data Availability

Data sharing does not apply to this article as it is a study protocol.

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
