# Peer review of "Effectiveness of 38% Silver Diamine Fluoride Application along with Atraumatic Restorative Treatment for Arresting Caries in Permanent Teeth When Compared to Atraumatic Restorative Treatment in Adults—Study Protocol for a Randomized Controlled Trial"

_mps, 2022, doi:10.3390/mps5060087_

Round 1
Reviewer 1 Report
The authors shall describe the state of the art of caries restorative treatments. Composite treatments shall be at least cited as the gold standard in restorative treatments of cavitated caries lesions.
In line 78 the authors mention preventive measures and soon after they describe the ART method.
After line 78, the authors could add a sentence like the following:
“Currently, the state of the art of caries treatment is represented by resin based composite restorations either in anterior or in posterior teeth.”
You could support this sentence adding the following reference for anteriors: Part 1 and Part 2 of the same paper: PMID: 25223143 and 25975063 and a third reference for posteriors: doi: 10.1016/j.jdent.2020.103494.
Line 144: please cite Alvear (reference 24) also here, not only at line 147. The reader may miss the reference for the SMART technique.
Considering the availability of Smartphones among population and dental practitioners it would be a nice idea to add to the protocol also pictures at baseline, 7days, 30days and 6 month.
A sample of 10-5 adults per group.
Line 198: The authors wrote: “without involving the pulp.” The sentence should be changed while a cavitated lesion may not be considered involving the pulp but, after manual excavation, it can show pulp involvement.
Line 199: The authors wrote: “Patients understand the significance of SDF and ART treatment to prevent tooth decay.”
The sentence should be rephrased as:
“Patients understand the significance of SDF and ART for the treatment and prevention of tooth decay.”
Line 202:
“The Patient will be enrolled” shall be changed to “The Patient will be enrolled”
Line 204:
“Patients with the absence of periodontal disease. “ While this is a clinical trial related to caries treatment/prevention I do not see any relationship with the presence of periodontal disease.
Please consider leaving or not this inclusion criteria
Author Response
Comment 1 : After line 78, the authors could add a sentence like the following:
“Currently, the state of the art of caries treatment is represented by resin based composite restorations either in anterior or in posterior teeth.”
You could support this sentence adding the following reference for anteriors: Part 1 and Part 2 of the same paper: PMID: 25223143 and 25975063 and a third reference for posteriors: doi: 10.1016/j.jdent.2020.103494.
Answer: Thank you for your comments
Thank you. The above-mentioned statement and the references have been included in the manuscript as advised in the Introduction. Paragraph 2
"Currently, the state of the art of caries treatment is represented by resin-based composite restorations either in anterior or posterior teeth”. (7,8,9)"
Comment 2
Line 144: please cite Alvear (reference 24) also here, not only at line 147. The reader may miss the reference for the SMART technique.
Answer: The statement has been cited as advised in the Introduction. Paragraph 8.
"There is a change in the reference number from 24 to 27 after the inclusion of references 7,8, and 9."
"The Silver Modified Atraumatic Restorative Technique, known as the SMART technique, is a modification of ART as stated in a case report, wherein the application of SDF was followed by conditioning with polyacrylic acid and restoration of the cavitated tooth with GIC (27)."
Comment 3: Considering the availability of Smartphones among population and dental practitioners it would be a nice idea to add to the protocol also pictures at baseline, 7days, 30 days and 6 month. A sample of 10-5 adults per group.
Answer:
A preoperative photograph of the selected tooth will be taken. A Photograph and a digital radiograph of the restored tooth will be taken after the procedure and at the end of the study.
Details regarding photographic records have been included in the protocol as advised in Methods. Section 2.7.
Comment 4.
Line 198: The authors wrote: “without involving the pulp.” The sentence should be changed while a cavitated lesion may not be considered involving the pulp but, after manual excavation, it can show pulp involvement.
Answer: Clinically only teeth with cavitation not involving the pulp will be included. If during the procedure, a pulp exposure occurs the patient will not be considered further for the study and will be replaced by another participant.
- "During the intervention, if there is inadvertent pulp exposure the patient will be excluded from the study."
This statement has been added to the exclusion criteria in Methods. Section 2.5.2
Comment 5.
Line 199: The authors wrote: “Patients understand the significance of SDF and ART treatment to prevent tooth decay.”
The sentence should be rephrased as:
“Patients understand the significance of SDF and ART for the treatment and prevention of tooth decay.”
Answer:
- Patients understand the significance of SDF and ART for the treatment and prevention of tooth decay.
The statement has been reframed as advised in Methods . Section 2.5.1
Comment 6.
Line 202: “The Patient will be enrolled” shall be changed to “The Patient will be enrolled”
Answer:
Patients will be enrolled irrespective of special needs if they are willing and able to comply with all the trial requirements.
The statement has been reframed as advised in Methods, Section 2.5.1
Comment 7 : Line 204: “Patients with the absence of periodontal disease. “ While this is a clinical trial related to caries treatment/prevention I do not see any relationship with the presence of periodontal disease.
Please consider leaving or not this inclusion criteria
Answer: While periodontal disease might not have direct interaction with SDF/ART treatment, the tooth could be compromised with a poor prognosis. Hence, we decided to exclude patients with periodontal disease.

Reviewer 2 Report
The protocol is very detailed. The only point that should be rethought is the age of the patients. From 18 to 65 years old is a large range. Please review this point.
Author Response
The protocol is very detailed. The only point that should be rethought is the age of the patients. From 18 to 65 years old is a large range. Please review this point.
Thanks for the comments. This range is included to ensure that effectiveness of the ART With SDF application across age group.
Reviewer 3 Report
The studied topic is interesting and the manuscript is well written.
Author Response
The studied topic is interesting and the manuscript is well written.
Thanks for comments